# Timekeeping in the hindbrain: a multi-oscillatory circadian centre in the mouse dorsal vagal complex

Lukasz Chrobok[1,2,7], Rebecca C. Northeast[1,7], Jihwan Myung 🄳 [3,4,5], Peter S. Cunningham[1], Cheryl Petit 🄳 [1] & Hugh D. Piggins[1,6✉]

Metabolic and cardiovascular processes controlled by the hindbrain exhibit 24 h rhythms, but the extent to which the hindbrain possesses endogenous circadian timekeeping is unresolved. Here we provide compelling evidence that genetic, neuronal, and vascular activities of the brainstem's dorsal vagal complex are subject to intrinsic circadian control with a crucial role for the connection between its components in regulating their rhythmic properties. Robust 24 h variation in clock gene expression in vivo and neuronal firing ex vivo were observed in the area postrema (AP) and nucleus of the solitary tract (NTS), together with enhanced nocturnal responsiveness to metabolic cues. Unexpectedly, we also find functional and molecular evidence for increased penetration of blood borne molecules into the NTS at night. Our findings reveal that the hindbrain houses a local network complex of neuronal and non-neuronal autonomous circadian oscillators, with clear implications for understanding local temporal control of physiology in the brainstem.

---

[1] Faculty of Biology, Medicine and Health, University of Manchester, Oxford Road, Manchester M13 9PT, UK. [2] Department of Neurophysiology and Chronobiology, Institute of Zoology and Biomedical Research, Jagiellonian University in Krakow, Gronostajowa Street 9, 30-387 Krakow, Poland. [3] Graduate Institute of Mind, Brain, and Consciousness, Taipei Medical University, No.172-1 Sec. 2 Keelung Road, Da'an District, Taipei 106, Taiwan. [4] Graduate Institute of Medical Sciences, College of Medicine, Taipei Medical University, No. 250, Wu-Hsing Street, Taipei 110, Taiwan. [5] Brain and Consciousness Research Centre, Taipei Medical University-Shuang Ho Hospital, Ministry of Health and Welfare, No. 291 Zhongzheng Road, Zhonghe District, New Taipei City 235, Taiwan. [6] School of Physiology, Pharmacology, and Neuroscience, Faculty of Life Sciences, University of Bristol, Bristol BS8 1TD, UK. [7] These authors contributed equally: Lukasz Chrobok, Rebecca C. Northeast. ✉email: hugh.piggins@bristol.ac.uk

Conventionally, the suprachiasmatic nuclei (SCN) are considered as the master circadian clock and while ambient light is its predominant environmental synchroniser, resultant circadian rhythms are also influenced by arousal and ingestive cues[1–5]. Cells of the SCN function as autonomous clocks as they contain a molecular clock of which *Period* (*Per1-2*) and *Cryptochrome* (*Cry1-2*) genes and their proteins (PER1-2/CRY1-2) are important negative feedback components[6]. The transcription and translation of these genes drive a 24 h cycle of protein accumulation and degradation. This leads to daily variation in SCN neuronal action potential discharge[7,8] and the promotion of clock cell synchrony. Through timed synaptic and/or paracrine signals, the SCN then orchestrates circadian variation in the brain and corresponding physiological outputs[5]. However, despite the apparent primacy of the SCN, accumulating evidence challenges this uni-clock model[4,9]. In SCN-less tissue explants from rodents bearing bioluminescence reporters of the molecular clock (e.g. PER2::LUC mice[10]), both autonomous and semi-autonomous rhythms of clock gene expression are observed in additional structures including the olfactory bulb[11], choroid plexus[12], habenula[13,14] and mediobasal hypothalamic nuclei[15] as well as peripheral tissues[16]. Thus, daily control in physiology and behaviour appears partially devolved to local clocks.

The hindbrain is a highly conserved multi-component structure critical for survival[17–19], yet the extent to which it displays intrinsic circadian timekeeping is unclear. Here we show that the dorsal vagal complex (DVC) of the brainstem contains a network of circadian oscillators that self-regulate to generate different dynamic qualities. This hindbrain complex functions as the main gateway for peripherally generated signals entering the brain and is correspondingly implicated in autonomic and metabolic control[20]. We determine the distinct oscillatory properties of two of its anatomical structures, the area postrema (AP), nucleus of the solitary tract (NTS), as well as non-neuronal components forming the ependymal cell layer of the fourth ventricle (4thVep) and the specialised glial barrier separating the NTS from the AP[21]. Neurons of the AP innervate the NTS[22–26] and we show that this pathway is important for coordinating phases between these structures. Further, we find that the AP and NTS also express day-night variations of electrical activity and neuronal responses to metabolically relevant signalling factors. At night, both genetic and functional studies show that blood borne molecules have enhanced access to the NTS. Together these studies reveal unexpectedly extensive systematic daily regulation of molecular, cellular, network, and vascular activities in the DVC.

## Results

### Anatomical components of the DVC exhibit robust oscillations of PER2::LUC.
It is unclear if the DVC contains intrinsic circadian oscillators and to address this, we cultured coronal brainstem slices and imaged PER2::LUC expression over 7 days ($n = 8$). Robust rhythms in bioluminescence were evident in the AP, NTS and in the 4thVep and typically measurable for 5–6 days in culture (Fig. 1a, Supplementary Fig. 1). Additionally, transient (one day) PER2::LUC expression was detected in the subjacent dorsal motor nucleus of the vagus (DMV), a key regulator of parasympathetic brain output that is under the control of NTS neurons (Supplementary Fig. 1). Notably, rhythms in PER2::LUC bioluminescence were anatomically limited to DVC explants containing the AP (Supplementary Fig. 2) and were absent in DVC sections rostral or caudal to the AP.

Both manual visual inspection (Fig. 1a–d) and automated image analysis (Fig. 1e, f) revealed that the AP, NTS and the 4thVep function as three independent circadian oscillators. A spatiotemporal sequence in PER2::LUC expression was evident

whereby the initial peak of the AP bioluminescence preceded that of the NTS by ~2 h ($p < 0.0001$), and the 4thVep by ~10 h ($p < 0.001$). Consequently, the rhythm in PER2::LUC in the 4thVep was in near antiphase to that of the AP (Fig. 1a–c, e, f). The circadian period of these bioluminescence rhythms was similar in the AP and NTS (~24.7 h and ~24.3 h respectively), but significantly shorter in the 4thVep (~22.9 h; compared with AP, $p < 0.05$) (Fig. 1d). Unbiased, grid-based whole-image analysis of bioluminescence also identified independent oscillatory clusters corresponding to these anatomical subregions in the DVC (Fig. 1e) and confirmed the inverted relationship between initial phase and period length, with slower oscillators being relatively advanced in phase (Fig. 1f). This unexpected relation suggests that in vivo, coordinated network coupling acts to stabilise phase alignment among these oscillators, whereas ex vivo, this coupling is absent and consequently, phase alignment among oscillators gradually collapses as each oscillator free-runs at its own intrinsic period, a process called detuning[27]. Simulations reveal that the degree of synchronisation can decay in a quadratic time course when there is no coupling[28]. However, we found that the AP and NTS ex vivo detuned differently from predicted time courses for complete decoupling. The AP sustained synchrony longer than the NTS (mean $R$, $p < 0.01$), and in both structures, the detuning time course revealed that some of the heterogenous coupling remained ex vivo (Supplementary Fig. 3). This suggests that in vivo, the coupling structure among oscillators is complex, with a subset of DVC oscillators preferentially connected.

Damping of the whole area PER2::LUC rhythms in the DVC can arise through reduced clock gene expression in single cells, progressive desynchronisation amongst these single cells (detuning) or a combination of the two. With manual visual inspection, putative single cell oscillators could be reliably discriminated in the AP and NTS (but not in the 4thVep), allowing individual assessment of their PER2::LUC expression (Fig. 1g). Single cell oscillators desynchronised visibly throughout the recording coincident with the damping of their respective networks (Fig. 1g,i and Supplementary Fig. 4A). Initially, the average phase peak of the oscillating AP cells was CT9.4 ± 0.2 h, while that of NTS cells occurred ~1.5 h later at CT10.91 ± 0.6 h (day 2, $p < 0.05$). Cells desynchronised at similar rates in both structures, but throughout the recordings, single cells in the AP were more synchronised than those in the NTS (Fig. 1i). Consistent with this, the distribution of phase differences varied between these structures (2-way ANOVA interaction: $p < 0.0001$) (Fig. 1j). The average period of single oscillating AP cells was slightly longer than that of NTS cells ($p < 0.05$), but less variable ($p < 0.05$), which potentially accounts for the reduced synchrony in the latter structure (Supplementary Fig. 4B, C).

Following 7 days in culture, whole area PER2::LUC rhythms in the DVC were significantly dampened and subsequent application of 10 μM forskolin ($n = 3$) re-initiated whole area and single cell oscillations for up to a further 14 days. This treatment transiently re-synchronised phasing in the AP and NTS (Supplementary Fig. 5A, C, D). However, by the 5th day following forskolin treatment, the initial spatiotemporal sequence seen during baseline recordings of the AP leading the NTS and the 4thVep had resumed (Supplementary Fig. 5B). This is further evidence that these oscillators can be phase synchronised, but that altered dynamics in coupling seen here ex vivo changes these phase relations. Detailed analysis of the PER2::LUC expression is reported in Supplementary Table 1.

### The DVC components exhibit cyclic clock gene expression in vivo.
Rhythmic clock gene expression in the DVC is not widely reported and we next assessed the in vivo *Per2* and *Bmal1*

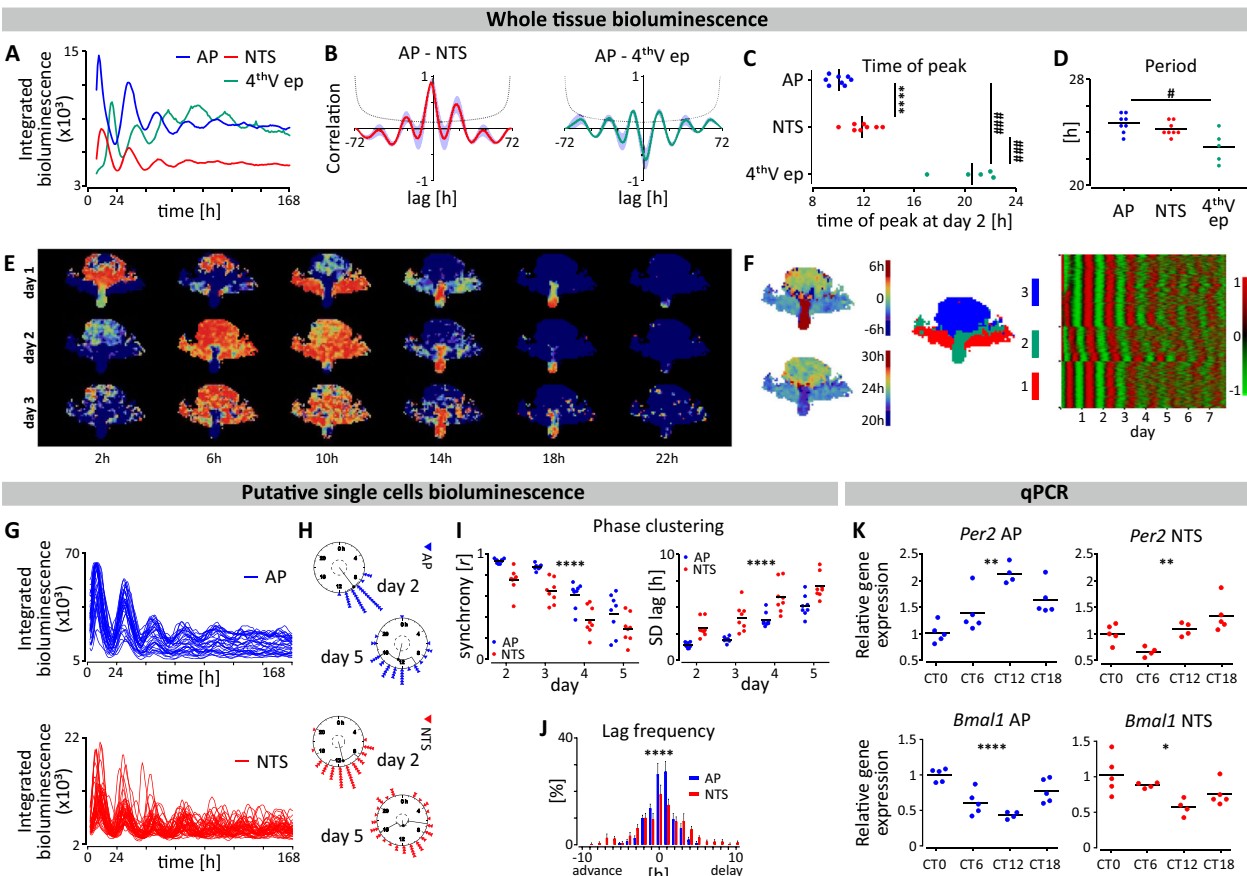

**Fig. 1 Spatiotemporal variation in _Per2_/PER2 in the AP and NTS. a** Example traces of PER2::LUC expression over 7 days in culture at the whole tissue level in the AP, NTS and 4thVep from one brainstem slice. **b** Cross-correlograms showing the ~2 h lag of NTS to AP (left panel; $n = 8$) and close to antiphasic relationship of 4thVep to AP (right panel; $n = 5$) over 72 h. **c** Altered times of peak PER2::LUC bioluminescence (AP vs NTS ****$p < 0.0001$, $n = 8$, $t$ test; 4thVep vs AP and NTS ###$p < 0.001$, $n = 5$, Tukey's test). **d** Periods of whole area structures of the brainstem (#$p < 0.05$, $n = 5$, Tukey's test). **e** Reconstructed images from grid-based, normalised PER2::LUC activities in the brainstem slice showing phase of spatial PER2::LUC expression every 4 h over 3 days. Warm colours code toward peak phase. **f** (from left to right) The average phase of the oscillation for the first 24 h in culture (upper left) and the period for the first 48 h (lower left), the map of clustering based on the correlation of time series (middle), and the raster plot of normalised time series of the oscillation listed in cluster groups (each row represents one grid) (right). The first cluster (_red_) approximates the anatomical position of NTS, the second one (_green_) the 4thVep, and the third cluster (_blue_) the AP (middle). **g** Traces of putative single cell PER2::LUC expression over 7 days in culture. **h** Rayleigh plots depicting phase clustering around CT 8.5 in AP and CT 11 in NTS at day 2, and subsequent desynchronisation of single cell oscillators at day 5. **i** Rayleigh $r$ value of individual cells for peaks 2-5 (left panel) and the SD of individual cell lags relative to the phase of the main structure for peaks 2-5 (right panel) for $n = 8$ cultures. (****$p < 0.0001$, two-way ANOVA RM). **J.** Frequency histogram of single cell lags relative to the phase of the whole structure (****$p < 0.0001$, two-way ANOVA Interaction). **k** In vivo core clock gene expression of _Per2_ and _Bmal1_ in the AP and NTS relative to CT0. (*$p < 0.05$, **$p < 0.01$, ****$p < 0.0001$, one-way ANOVA).

transcript expression in the AP and NTS separately at 6 h intervals over the circadian cycle, beginning at the onset of the circadian day (CT0). Both _Per2_ and _Bmal1_ expression in the AP and NTS varied significantly over these time points (AP: $p < 0.01$ and $p < 0.0001$; NTS: $p < 0.01$ and $p < 0.05$, respectively) (Fig. 1k and Supplementary Tab. 2), with a delayed peak in _Per2_ in the NTS consistent with the phasing observed in PER2::LUC rhythms. Thus, in the mouse NTS and AP, molecular clock components also vary in expression over 24 h in vivo.

**Daily variation in DVC neuronal activity and electrical connectivity ex vivo.** For effective communication of circadian information, the molecular clock drives SCN neurons to increase spike frequency during the day and to lower it at night[8,29]. To assess whether DVC neurons are similarly coordinated over 24 h, we simultaneously recorded spontaneous multi-unit activity (sMUA) throughout coronal DVC brain

slices. In recordings made continuously for up to 26 h ($n = 4$), clear 24 h variation was seen in all slices (Fig. 2a). The waveform of this electrical profile was significantly wider in the AP than the NTS ($p < 0.0001$; Fig. 2b), but the time of peak in sMUA occurred late in the projected day for both structures (ZT9.6 ± 0.6 h vs ZT9.6 ± 0.4 h; Fig. 2c). Acute recordings of sMUA made at two opposing time points for day (ZT3-4, $n = 22$ slices; $n = 11$ mice) and night (ZT15-16, $n = 22$ slices; $n = 11$ mice) confirmed this. In both the AP and NTS, sMUA was detected on a larger proportion of electrodes during the night than the day (AP: $p < 0.05$; NTS: $p < 0.0001$; Fig. 2d, Supplementary Tab. 3) and the frequency of sMUA was also higher at night compared to the day (AP: $p < 0.01$; NTS: $p < 0.0001$; Fig. 2e,f and Supplementary Table 3). Collectively, this indicates that similar to the SCN, rhythmic clock gene activity in the brainstem DVC can convey timekeeping information through day-night variation in the activity of AP and NTS neurons.

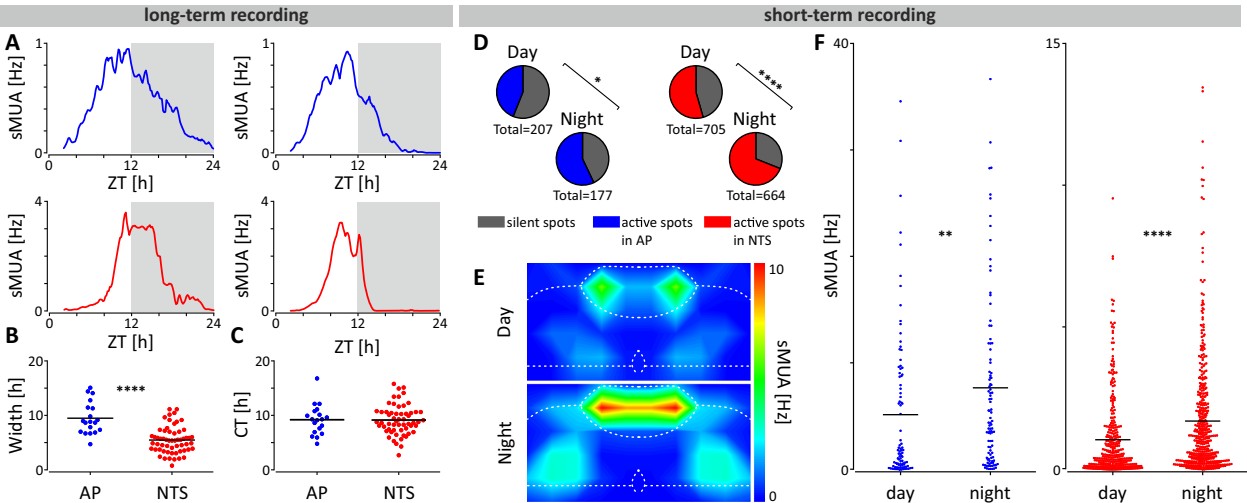

**Fig. 2 AP and NTS neurons exhibit robust 24 h and projected day-night variation in electrical activity. a** Example long-term recording profiles of spontaneous multi-unit activity (sMUA; bin = 600 s) in the AP (*in blue*) and NTS (*in red*) showing the peak of electrical activity at late projected day/early night. Grey boxes depict the projected dark phase. All recordings were started at ZT2. **b** The width of the 24 h sMUA profiles for individual recording sites (*n* = 4 slices from 3 mice) in the AP and NTS taken at 50% of the peak amplitude (****p < 0.0001, *t* test). **c** The projected ZT of the peak of sMUA of each electrode in the AP and NTS with an average of ~ZT9.5 for both structures. **d** For the short-term recording protocol (ZT3-4 vs ZT15-16) sMUA was detected at greater proportion of electrodes in the AP and NTS during the projected night (active spots; blue for AP and red for NTS) (*p < 0.05, ****p < 0.0001, Fisher's tests). **e** Mirror heatmaps coding the levels of sMUA in the DVC complex at projected early day (above) and early night (below) obtained from short-term recordings. **f** sMUA from single recording electrodes localised in the AP (*in blue*; *n* = 192) and NTS (*in red*; *n* = 842); sMUA was significantly elevated early night in both structures (**p < 0.01, ****p < 0.0001, *t* tests).

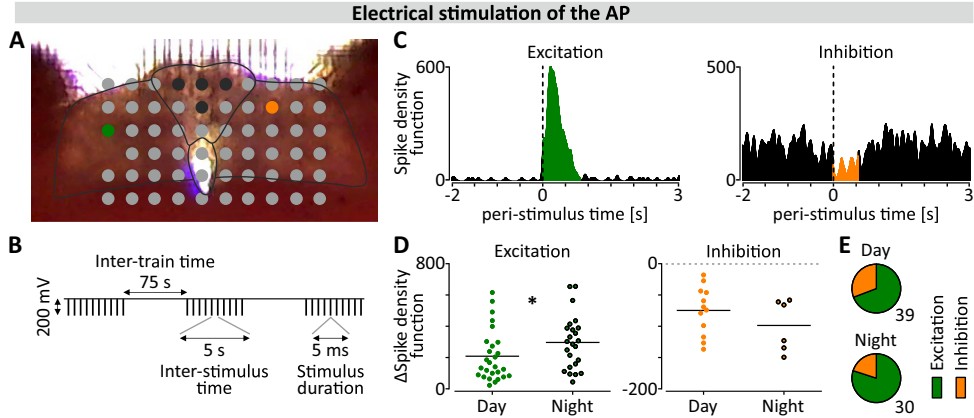

**Fig. 3 Daily changes in AP to NTS communication. a** Photograph showing the position of the brainstem section on the electrode array; black represents stimulation sites, green and orange represent electrodes where the example excitatory and inhibitory responses (shown in **c**) were recorded. Black lines delineate structures: the AP and the borders of DVC. **b** Schematic representation of the stimulation protocol. **c** Representative excitatory and inhibitory responses in the NTS shown in modified peri-stimulus time histograms (PSTHs) representing the sum of spike density function around 30 stimuli. **d** Amplitudes of nocturnal excitations were significantly higher than those recorded during the day (*p = 0.042, Mann–Whitney test), but inhibitory responses did not show this day-to-night increase (p = 0.336, Mann Whitney test). **e** Proportion of excitations (*in green*) to inhibitions (*in orange*) recorded at day and night.

**Separation of independent DVC oscillators alters circadian dynamics in the NTS.** Neuronal connections from the AP to the left and right NTS can be maintained in coronal brainstem slices ex vivo[22] and consistent with this, we found that electrical stimulation of electrodes in the AP evoked both activations and suppressions at recording sites throughout the whole NTS (*n* = 8; Fig. 3a–c). The magnitude of the activations increased at night (ZT18 vs ZT6, *p* < 0.05; Fig. 3d), while that of the suppressions did not (*p* = 0.336; Fig. 3e). This indicates that AP to NTS communication varies across the circadian cycle and we next asked whether mechanical separation of the NTS from the AP influences the circadian properties of these structures. Therefore, we took brainstem explants from PER2::LUC mice for

bioluminescence imaging experiments and unilaterally severed connections between the NTS and the AP, leaving the connections on the contralateral side intact (*n* = 5; Fig. 4a). Recording simultaneously across the coronal brainstem section, bioluminescence rhythms in whole area and single cell oscillators were still reliably observed in all three regions of interest (ROIs): the AP, the NTS connected to AP (NTSc) and the entirely disconnected NTS (NTSd) (Fig. 4b). Oscillations sustained in the whole NTSd demonstrated the same rate of relative oscillatory damping as in the NTSc (*p* = 0.417; Fig. 4c). However, the period of the NTSd rhythm (21.53 ± 0.6 h) was significantly shorter than both those in the AP (24.47 ± 0.3 h, *p* < 0.01) and the NTSc (24.60 ± 0.7 h, *p* < 0.01). The period of the NTSc did not differ

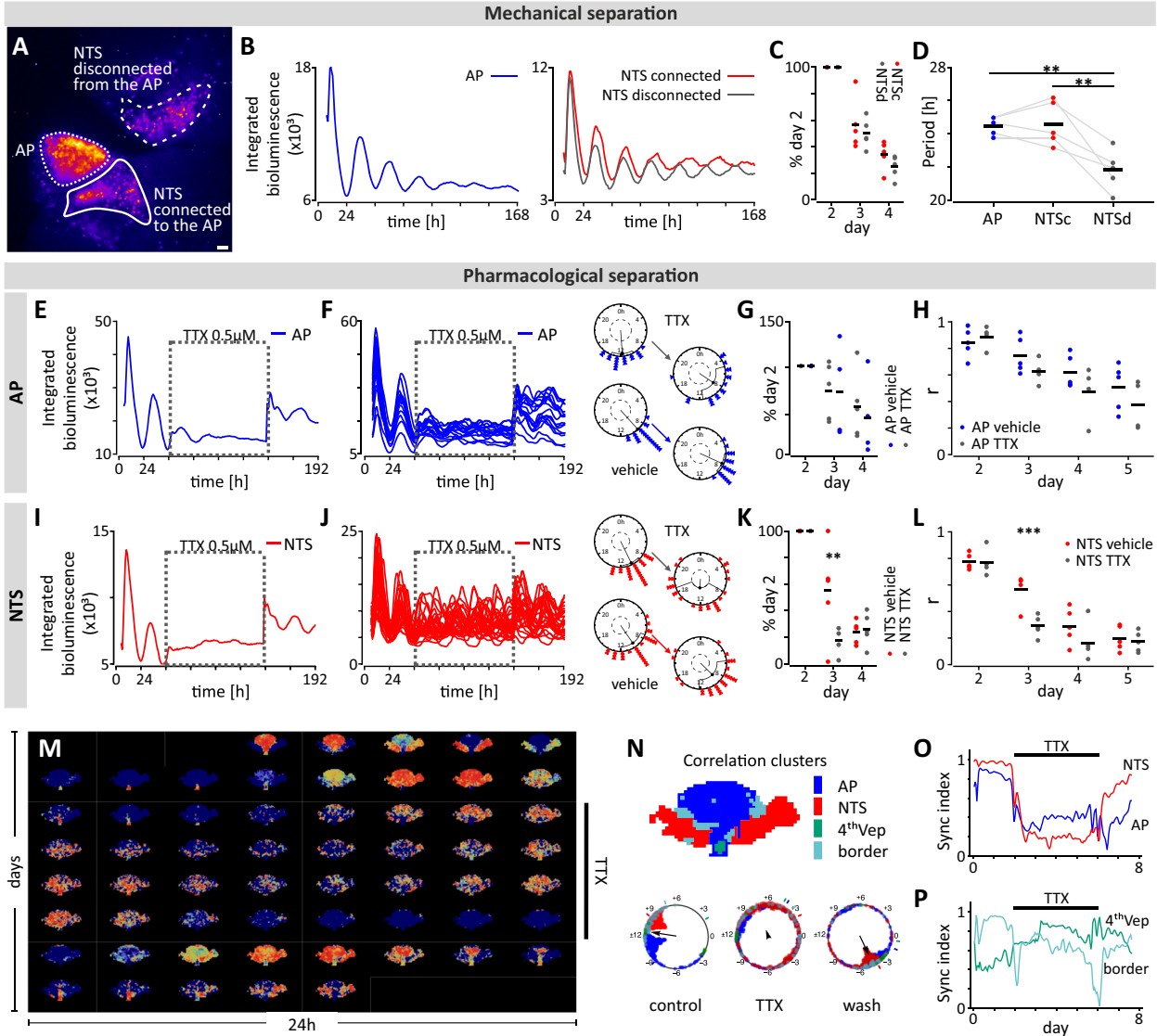

**Fig. 4 Mechanical and pharmacological disconnection reveals the circadian properties of DVC oscillators. a** False-coloured bioluminescence image of DVC explant with the unilateral NTS disconnected from the AP. Borders of three regions of interest (ROIs) were delineated and overlaid. White bar depicts 100 μm. **b** Example traces of wholearea PER2::LUC expression for the AP (*in blue*), NTS connected (NTSc, *in red*) and NTS disconnected from the AP (NTSd, *in grey*) from one hindbrain slice. **c** Damping rate, shown as relative amplitude to day 2, did not vary between NTSc and NTSd ($p = 0.4172$, two-way ANOVA RM). **d** Period of NTSd PER2::LUC rhythms was shorter than the AP ($n = 5$, **$p = 0.0061$, Tukey's test) or NTSc ($n = 5$, **$p = 0.0045$, Tukey's test). **e,i** Example PER2::LUC bioluminescence traces showing effects of tetrodotoxin (TTX, 0.5 μM) (AP – *blue*, NTS – *red*) and TTX washout (*dotted box*). **f, j** Single cell bioluminescence traces during TTX treatment with example Rayleigh plots of one day pre- and one day post-treatment (day 2 and 4, respectively) for both structures. **g, k** TTX had no effect on whole tissue damping rate in the AP ($p = 0.8043$, two-way ANOVA RM), but did in the NTS (day 3: **$p = 0.0046$, Sidak's test). **h, l** Rayleigh $r$ value showing synchrony amongst single cell oscillators from day 2 to 5. TTX did not reduce synchrony in the AP ($p = 0.2380$, two-way ANOVA RM), but desynchronised NTS cells (day 3: ***$p = 0.0007$, Sidak's test). **m** Reconstructed normalised PER2::LUC time-lapse images show loss of the spatial phase organisation under TTX. **n** (upper) Correlation clusters maintain a canonical phase organisation before TTX. (lower) The phase relation between the AP and NTS under control conditions is lost with TTX elicited desynchrony. NTS regains synchronisation after wash. The small bars external to the circle indicate average phases of respective clusters. Black arrow direction indicates the average phase of all oscillators, and its length indicates the degree of synchronisation. **o** Neuronal constituents of the NTS and AP respond rapidly to TTX and desynchronise. **p** TTX has no effect on the synchrony of the non-neuronal constituents of the 4thVep and the border cluster. The sync index is quantified by the Kuramoto order parameter.

from that of the AP ($p = 0.983$; Fig. 4d). These observations indicate that severing of AP to NTS connections unmasks the intrinsic short period rhythms of the adjacent NTS and alters phasing, with the rhythm in the NTSd becoming antiphasic over time to the rhythms of the AP and NTSc (Supplementary Fig. 6A).

The disconnection of the NTS from the AP also compromises the ability of the single cell NTS oscillators to maintain synchrony. Single cell rhythms in the NTSd desynchronised more quickly than single cells in the NTSc ($p < 0.05$, two-way ANOVA interaction; Supplementary Fig. 6B, C). Thus, anatomical and functional connections with the AP slow the oscillators in the NTS and promote single cell synchrony in this structure (see Supplementary Table 4 for statistical analysis).

Since neuronal signals from the AP regulate key rhythmic parameters of the NTS oscillator, we next examined the

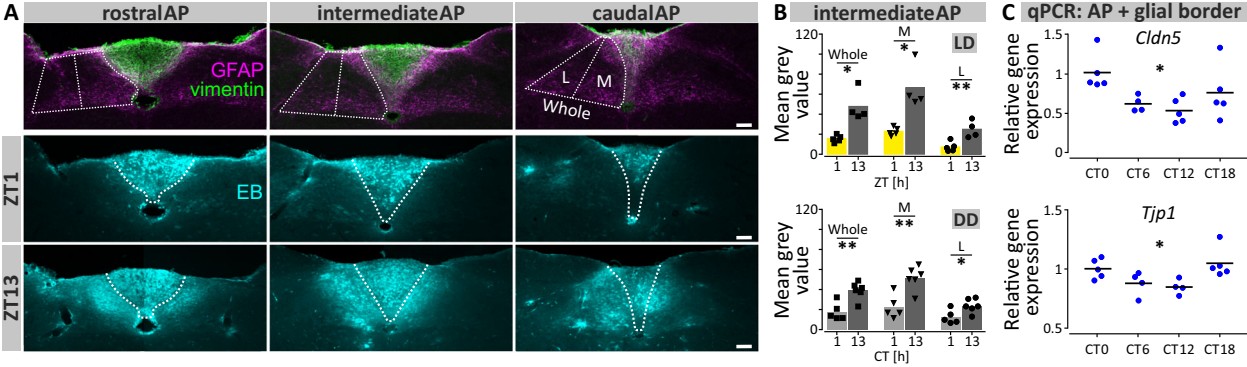

**Fig. 5 Increased permeability to Evans Blue dye in the NTS at early night. a** Representative images of the rostral-caudal DVC showing the merge of anti-vimentin and anti-GFAP staining (top row) for identification of the AP/NTS glial border along with annotations of the ROIs of the NTS taken for subsequent analysis. Evans Blue dye (EB) fluorescence imaging (middle & bottom rows) with added annotation of the AP/NTS border at ZT1 vs ZT13; note the increased penetration of EB into the adjacent NTS at ZT13. White bar depicts 100 μm. **b** Intensity of EB staining in the whole, medial and lateral areas of the NTS for the intermediate level sample slice during LD (top panel) or DD (bottom panel) (*$p < 0.05$, **$p < 0.01$, t tests or Mann–Whitney tests). **c** Gene expression relative to CT of the tight junction proteins Claudin-5 (Cldn-5) and Zona occludens-1 (Tjp-1) in the AP/glial border over 24 h (*$p < 0.05$, one-way ANOVAs). L lateral, M medial.

contribution of action potential-dependent communication in the maintenance of rhythmic PER2::LUC expression in the AP and NTS. To this end, we applied tetrodotoxin (TTX, 0.5 μM; a voltage-gated sodium channel blocker) to the culture media after 48 h of baseline recording ($n = 5$ brainstem slices) and manually assessed PER2::LUC bioluminescence. This visual inspection revealed that at both whole tissue and single cell level, TTX application had little effect in the AP ($n = 5$ slices; Fig. 4e, f; Supplementary Fig. 7A Supplementary Table 5) with neither damping of oscillations ($p = 0.804$; Fig. 4g) nor single cell synchrony altered ($p = 0.238$; Fig. 4h). By contrast, in the NTS, TTX rapidly accelerated the damping of whole tissue rhythms ($p < 0.01$; Fig. 4i, Supplementary Fig. 7B), and desynchronised single cell oscillations ($p < 0.001$; Fig. 4j, k, l; Supplementary Fig. 7B, C, Supplementary Table 5). Unbiased grid-based analysis confirmed these assessments (Fig. 4m, n) with the NTS cluster prominently desynchronised in response to TTX, whereas the synchrony index of the AP less obviously reduced (Fig. 4o). The cluster corresponding to the 4thVep was unaffected by TTX (Fig. 4p). This result was expected, since this region contains non-neuronal ependymal cells that do not use action potential-dependent synaptic communication. Surprisingly, this automated analysis also identified a fourth correlation cluster that approximated the border between the AP and NTS that was not affected by the TTX (Fig. 4n, p). This area contains non-neuronal cells forming the specialised glial barrier between the AP and NTS[30] and thus synchrony here was also unaltered by blocking action potential-dependent communication.

These experiments reveal that both AP input as well as action potential-dependent communication shape the parameters of the NTS circadian oscillation. In addition, these findings allude to a TTX-resistant oscillator localised to the region of the glial barrier between the AP and NTS.

**Daily variation in tight junction affects permeability through AP/NTS glial border.** Our findings of circadian variation in neuronal communication from the AP to NTS and the presence of a rhythmic cluster at the AP/NTS border led us to assess whether this glial border demonstrates circadian changes in its function. The intermediate filament proteins, vimentin and glial fibrillary acidic protein (GFAP) are expressed by astrocytes and immunostaining for these delineates the interface between the AP and NTS (Fig. 5a, top panel). This area contains tight junction

proteins which function to regulate the diffusion of substances from the AP to the NTS parenchyma, thereby forming a highly specialised barrier separating the NTS from this circumventricular organ (CVO)[31,32]. To test temporal variation in the permeability of this barrier, we perfused mice with Evans Blue (EB) dye at early day or early night time points under LD and DD conditions (ZT1/CT1 and ZT13/CT13, respectively). During the early day, EB dye was mostly confined to the AP with minimal presence in the subjacent and adjacent NTS (Fig. 5a, middle panel). However, at early night EB dye was present in the medial and, to a lesser extent, the lateral aspects of the NTS as well as throughout the AP (Fig. 5a, bottom panel). This temporal variation in EB staining was seen under both LD and DD conditions, indicating the endogenous circadian regulation of the permeability of the AP-NTS border (Fig. 5b, Supplementary Fig. 8 and Supplementary Tab. 6). Since our micro-dissected AP samples contained this boundary area (Supplementary Fig. 9), and because tight junction components in the kidney and gastro-intestinal tract are under circadian control[33,34], we assessed temporal expression of genes coding for the tight junction components zona occludens-1 (Tjp1) and claudin-5 (Cdn5). Expression of Tjp1 ($p < 0.01$) and Cdn5 ($p < 0.05$) varied over the circadian cycle, with the lowest expression in both transcripts occurring around the onset of circadian night (CT12; Fig. 5c, Supplementary Table 2) coincident with the high permeability to EB dye. Examination of other sensory CVOs, the subfornical organ (SFO) and vascular organ of lamina terminalis (OVLT; Supplementary Fig. 10) revealed that EB dye was contained within these CVOs at both early day and early night time points. Thus, the nocturnal increase in permeability to EB is restricted to the DVC, raising the possibility that the NTS can function as a CVO-like structure during the night. Taken together, our results implicate a role for locally autonomous circadian clock regulation in tight junction protein expression to allow for daily changes in permeability to blood borne molecules and/or other humoural factors. This indicates that the circumventricular organ characteristics of the DVC expand at night.

**Day-night alterations in the NTS integration of metabolic signals.** Since permeability to humoural factors increased at night, we next evaluated if NTS neurons also exhibited elevated nocturnal responsiveness to interoceptive metabolic signals. This includes factors that reduce and/or terminate ingestive behaviours

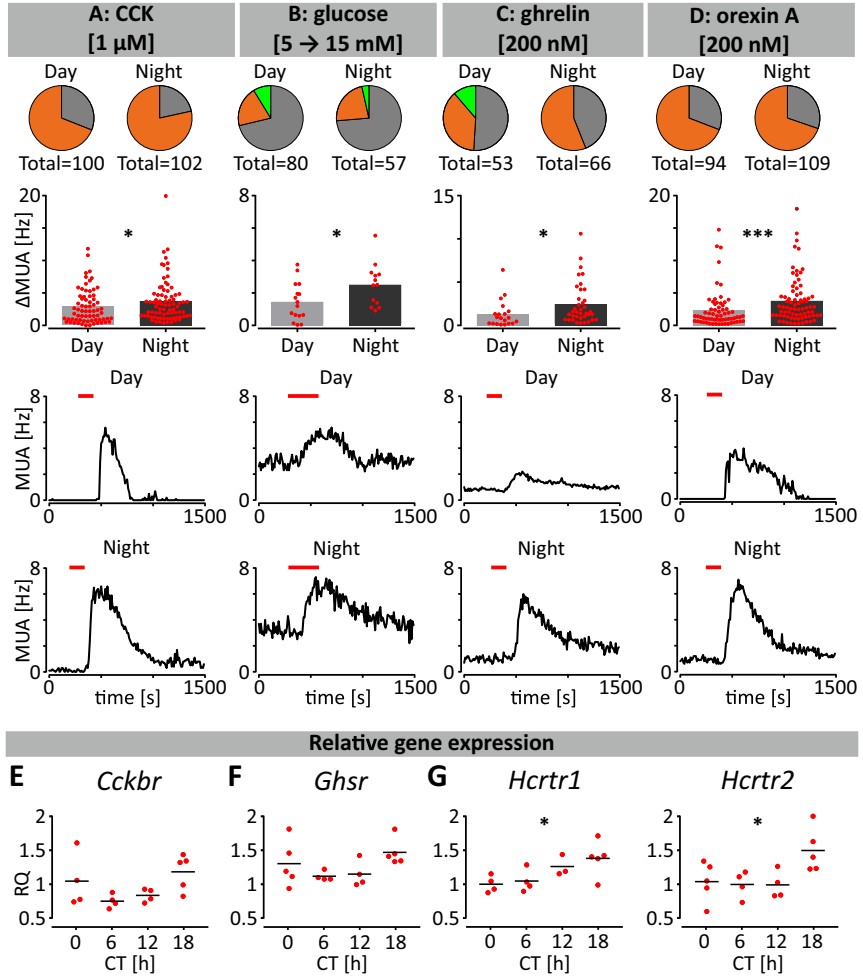

**Fig. 6 Day-night variation in responsiveness of NTS neurons to metabolic factors. a–d** Multi-electrode array recordings conducted during the day (ZT4-6) or night (ZT16-18) reveal that NTS neurons increased nocturnal responsiveness to (**a**) CCK, (**b**) elevated glucose, (**c**) ghrelin, and (**d**) orexin A. Pie charts represent the proportion of NTS recording locations responding through activation (*orange*), inhibition (*light green*) or no change in firing activity (*grey*). Bars represent the amplitude of drug-evoked excitation or inhibition (*$p < 0.05$, ***$p < 0.001$ *t* test or Mann–Whitney *t*est). Multi-unit activity traces showing representative responses to drug application during the day or night are presented below. **e–g** Temporal variation in gene expression for metabolic receptors in the NTS sampled at CT0, 6, 12, and 18. Data are plotted relative to the values at CT0. Orexin receptor gene expression (*Hcrtr1* and *Hcrtr2*) varied across these time points. *$p < 0.05$, one-way ANOVAs or Kruskal–Wallis tests.

such as raised concentrations of blood glucose and the gastrointestinal hormone cholecystokinin (CCK)[35–37]. We found that CCK application evoked activations from the majority of NTS electrodes and that the amplitude of these activations was significantly increased at night ($p < 0.05$; Fig. 6a). Exposure to elevated glucose elicited activations in NTS MUA from ~20% of recording sites during the day and night, with the amplitude of response significantly increased at night ($p < 0.05$, Fig. 6b). CCK signals through the CCKB receptor (CCKBR), and to assess if this receptor could vary in availability in the NTS, we measured *Cckbr* expression in this structure and found a trend in its daily expression with the lowest level at CT6, and the highest at CT18 ($p = 0.087$, Fig. 6e). This suggests that increased night-time *Cckbr* expression could contribute to enhanced nocturnal responsiveness of NTS neurons to CCK.

NTS neurons are also responsive to chemical cues that promote arousal and ingestive behaviour, including the hypothalamic neuropeptide orexin A (OXA) and the gastrointestinal hormone ghrelin[38–40]. Ghrelin application elicited activations both day and night in the NTS MUA, with the amplitude of the response increased at night ($p < 0.05$, Fig. 6c). Suppression of firing to

ghrelin was infrequently observed with no day-night differences in these response characteristics. OXA enhanced MUA from ~70% of NTS recording sites, both day and night, with an increased amplitude in the nocturnal response ($p < 0.001$, Fig. 6d). Ghrelin signals via the ghrelin receptor (Ghsr) while orexin acts via the HCRTR1 and HCRTR2 receptors, and to investigate whether NTS expression of these receptors varied across the circadian cycle, we examined *Ghsr* and *Hcrtr1-2* expression in the NTS of mice culled at 4 timepoints in constant dark. No significant temporal changes were observed in *Ghsr* expression in the NTS ($p = 0.132$, Fig. 6f). However, expression of transcripts for orexin receptors (*Hcrtr1* and *Hcrtr2*) was significantly elevated during the night, with the highest level at CT18 ($p < 0.05$, Fig. 6g). Statistical analysis of qPCR measurements and drug treatments are presented in Supplementary Tables 2 and 7, respectively. Therefore, NTS neurons exhibit elevated nocturnal responsiveness to metabolic and arousal cues as well as gene expression of some of their cognate receptors. This suggests that one function of circadian processes in the NTS is to enhance its night-time sensitivity and processing of signals associated with metabolic and arousal state.

## Discussion

Here, we provide compelling evidence that the DVC contains a network of neuronal and non-neuronal components capable of independent circadian oscillations. These can drive daily variation in neuronal firing rate and day-night alterations in both responsiveness and potential exposure to metabolic cues. Additionally, we identify complex spatiotemporal dynamics by which the NTS can be synchronised to the rhythm of the AP. This process is dependent on intact connectivity between the two structures, the removal of which reveals the faster intrinsic oscillation of the NTS. Further, under constant dark conditions in vivo, we find that expression of circadian clock and tight junction genes in these structures varies over 24 h, with increased permeability to blood borne signals during the early night when expression of the core clock gene *Per2* is high and that of the tight junction components *Tjp1* and *Cldn5* low. Daily variation in receptor genes for metabolically relevant ligands was also detected in vivo in the NTS. Therefore, our findings reveal widespread circadian change in molecular, cellular, electrophysiological, and vascular activities within the brainstem and implicate intrinsic circadian timekeeping as a source of this temporal modulation.

Our in vivo study confirms the daily variation of *Per2* and *Bmal1* expression in the NTS[41,42] and extends this to show unexpectedly robust clock gene expression in the AP. Real-time ex vivo PER2::LUC imaging of the DVC revealed the presence of three independent hindbrain oscillators; the AP, NTS and 4thVep, with possibility of a fourth at the border between the AP and NTS. We found that the oscillatory properties of the AP were more robust than those for the NTS, presumably due to the increased synchrony observed between its single cell oscillators. Initially, PER2::LUC oscillations in all three hindbrain centres were sustained for up to 5–6 days and were reinstated by forskolin application, enabling robust oscillations that persisted for up to a further two weeks in culture. Interestingly, similar to findings in the subregions of the SCN[43–45], we observed a spatiotemporal wave in *Per2* expression across the hindbrain centres, originating and initially peaking first in the AP, before peaking some 2 h later in the NTS. In contrast, PER2::LUC in the 4thVep peaked in near antiphase to the AP, and subsequently oscillated with a consistently shorter period. Both the shorter period and anti-phasic relationship of the ependymal oscillation in *Per2* expression to neuronal rhythms are concordant with similar observations in the choroid plexus[12,46] and in other central regions[15]. Similar to SCN explants, the period of PER2::LUC oscillations in cultured DVC explants is longer than 24 h and thus does not match with the typical mouse endogenous locomotor rhythm (~23.7 h)[47]. Interestingly, SCN periodicity can be shortened by choroid plexus-derived diffusible factors to resemble that of whole animal behaviour[12]. Therefore, given the proximity of the fourth ventricle, oscillating ependyma and choroid plexus possibly modulate the properties of the AP and NTS circadian oscillations in vivo.

In addition to the rhythmic transcription of core clock genes, a key expression of SCN timekeeping activity is its robust 24 h variation in neuronal excitability, resulting in a higher firing rate of clock cells during the day[29]. These rhythmic changes in the electrical activity are necessary to communicate circadian phase amongst single cell SCN oscillators as well as to the rest of the brain and body[8]. Our study shows that neurons of the AP and NTS also express such rhythms. Long-term sMUA recordings showed firing rate peaking late in the day for both structures, but with the rhythm in firing rate being more synchronised between recording sites in the AP than in NTS. This observation further supports the interpretation that the AP cellular oscillators are more tightly coupled than those in the NTS. Detuning analysis showed that the NTS is more susceptible to external input,

possibly including signals from the AP. Shorter-term recordings from mice culled at opposing times of day confirmed that early night neuronal activity is higher than that at early day in both structures and is therefore not a consequence of slice deterioration during prolonged sMUA recordings. These observations indicate that peak neuronal activity in the DVC is phase delayed by ~3 h relative to the SCN, which exhibits maximal electrical activity during the middle of the behaviourally quiescent circadian day (ZT6.7[48]). We propose that the increased late day/early night activity of brainstem neurons anticipates the behavioural active phase and thus contributes to the animals' preparedness for imminent elevated metabolic demands and autonomic tone of the circadian night.

Our findings indicate that electrical activation of the AP evokes mostly excitatory responses from NTS neurons, and that the magnitude of this elicited activity is increased at night. Physical severing of this anatomical connection reveals the NTS to have an intrinsic short period such that it drifts out of phase with the AP. Further, the NTS that remains unilaterally connected to the AP expresses a slightly longer period than observed when the NTS is bilaterally connected to the AP. Thus, altering AP-NTS connectivity unmasks unusual coupling relationships among DVC oscillators. The observed phase reorganization ex vivo suggests similarities with the SCN where both phase-attractive and phase-repulsive coupling functions to organise molecular timekeeping between different SCN subregions, although the exact molecular and cellular mechanisms for the complex coupling remain to be clarified[49]. Nonetheless, the current findings demonstrate that spontaneous self-regulation is possible in the network of DVC oscillators and that the AP acts as a key driver for the maintenance of phasing and period in NTS.

Consistent with this, blockade of action potential dependent communication with TTX desynchronised NTS single cell oscillators to a far greater extent than did AP cells. This indicates that synchrony amongst cellular oscillators in the NTS relies on action potential dependent communication to a similar extent as the neonatal SCN[44,50], the subfornical organ and the vascular organ of the lamina terminals[51]. By contrast, TTX insensitivity in the AP resembles that of the arcuate and dorsomedial hypothalamic nuclei[15]. Such heterogeneity in the response of DVC oscillators to TTX could arise through differential expression of TTX-insensitive mechanisms of communication such as gap junctions and/or involvement of non-neuronal cells. Glial cells are important to the circadian function of the SCN as well as other non-neuronal tissues such as the choroid plexus[12,46,52,53] and ventricular ependymal layers[15] (and this study) which also exhibit robust circadian oscillation in clock gene expression. Intriguingly, automated image analysis also identified TTX-resistant oscillations in the region corresponding to the glial border between the AP and NTS. This suggests differential involvement of non-neuronal cells in subregions of circadian oscillators in the DVC.

Emerging evidence from invertebrates[54] and vertebrates[55] implicates circadian processes in regulating daily variation in the permeability of the blood-brain barrier as well as tissue barriers in the cornea[56], kidney[33] and large intestine[34]. Such changes are due to daily variation in the components of tight junctions[33,34], gap junctions[54], and functionality of pericytes[55]. Our findings of 24 h variation in the expression of zona occludens-1 and claudin-5, along with increased nocturnal permeability of the AP/NTS border to EB, provide further evidence that biological barrier function is subject to circadian regulation. Evans blue dye has a molecular weight of 961 Da[57], therefore indicating that the AP/NTS border is directly permeable to small circulatory molecules of at least a similar mass. This could arise through neuronal activity dependent changes in gene expression or direct actions of molecular clock proteins on the promotor regions of tight

junction genes[34,58]. In other studies conducted during the early day, EB was confined to the AP only and was not detected in the brain parenchyma[59,60]. The increased night-time permeability at the AP/NTS border is anatomically discrete as it was not detected at other CVO sites such as the SFO and OVLT. This suggests that there is a selective increase in the exposure of the brainstem to blood borne signals into the brainstem at night. This could enhance brainstem monitoring of the internal milieu as well as increasing the influence of peripheral physiology on central homeostatic mechanisms. Moreover, this places the NTS as a CVO-like structure that functions to directly probe blood borne substances at a temporally restricted nocturnal phase of the circadian cycle.

The NTS functions as a brainstem hub and processes a plethora of information pertinent to metabolic state[61]. We find that responses to both anorexigenic and orexigenic factors were elevated during the night. Anorexigenic cholecystokinin (CCK) evoked larger responses during the dark phase, concordant with elevated nocturnal expression of CCK receptor b. Similarly, a rise in glucose concentration that mimics its postprandial elevation in vivo elicited an increased excitatory response in NTS neurons at night. Orexigenic ghrelin and orexin A (OXA) more readily activated NTS neurons during the night phase, concomitant with the highest expression of the orexin receptors 1–2. Changes in electrical responsiveness to metabolic signals may arise from the general elevation in nocturnal excitability, differential receptor availability at different times of the day-night cycle, or a combination of these. These time of day differences have clear implications for our understanding of NTS physiology. The vast majority of electrophysiological investigations of the NTS are conducted at one phase of the 24 h cycle or just during the lights-on phase (circadian time not stated)[36–39] when responses to some ligands may be difficult to detect and/or measure, leading to inaccurate conclusions of their actions on the NTS. Overall, our findings expand the repertoire of circadian structures, such as the SCN, whose neuronal responses to signalling factors are shaped by the molecular circadian clock[62].

Experiments presented here determine key daily and circadian rhythm properties of three distinct populations of DVC cells and show for the first time rhythmic variation in neuronal and vascular activities in the brainstem. Further studies determining the glia and neuronal subtype as well as transcriptional repertoire of brainstem clock cells will illuminate the broader physiological relevance of this putative circadian centre in the hindbrain. These will enable the application of animal models (e.g. Bmal1$^{flox/flox}$ mice) and the development of tools to specifically target clock gene expressing cells in delineated populations of the DVC. Through such approaches, it will be possible to directly interrogate causal relationships between the molecular clock and rhythms in function of these three interacting regions of the brainstem.

In conclusion, we show that the DVC is a multi-oscillatory centre, where circadian mechanisms exert an unexpectedly prominent influence on molecular and cellular activity and function. The presence of a well-structured endogenous timekeeping centre in the hindbrain provides a new basis for circadian variation in the central processing of interoceptive and arousal cues as well as for regulating peripheral physiology via the autonomic nervous system.

## Methods

**Animals**. All mice were kept under standard lighting conditions of 12:12 h light - dark (LD) cycles, unless stated otherwise, with ad libitum availability of food and water. C57BL6J mice were provided by Charles River, Kent UK, all other genotypes (details below) were bred in house by the University of Manchester Biological Services Facility. All procedures and experiments were carried out with approval by the Research Ethics committee of the University of Manchester and in keeping with the UK Animal (Scientific Procedures) Act 1986. Experimental procedures were designed to minimize the number of animals used and their suffering. Night vision goggles (Cobra, UK) were used to prevent animal exposure to light during procedures carried out during the dark phase or constant dark conditions.

**Bioluminescence reporter activity**. *Tissue preparation*: A total of 21 adult male *mPer2Luc* knock-in (PER2::LUC) mice[10] between the ages of 10–20 weeks old were used for all bioluminescent imaging experiments. 250 µm thick coronal brain slices −7.20 to −7.26 mm from bregma[63] were cut in ice cold Hank's Balanced Salt Solution (HBSS; Sigma, Germany) supplemented with 1 mg/ml penicillin-streptomycin (Gibco Invitrogen Ltd, UK) and 0.01 M HEPES (Sigma) using a vibroslicer (Camden Instruments, UK). The DVC was dissected using a scalpel and prepared for culture. For the disconnection study, the unilateral NTS was cut out from the complex and further cultured in the same dish. The explant tissue was then transferred to 30 mm Millicell cell culture inserts (Merck, Germany) in glass-coverslip sealed Fluorodish culture dishes (World Precision Instruments Ltd., USA) with sterile culture medium (Dulbecco's Modified Eagle's Medium; DMEM, Sigma) supplemented with 0.1 mM luciferin (Promega, USA), B27 (Gibco Invitrogen Ltd, USA), 1 mg/ml penicillin-streptomycin (Gibco Invitrogen Ltd), 10 mM HEPES (Sigma) and 3.5 g/L D-glucose (Sigma).

*Data acquisition*: Images were taken using the Olympus Luminoview LV200 (Olympus, Japan) fitted with a cooled Hamamatsu ImageEM C900-13 EM-CCD camera and a 20 × 0.4 NA Plan Apo objective (Olympus, Japan) on a heated stage kept at 37 °C. Exposure time was 60 min and gain was kept constant throughout. 10 µM forskolin (Sigma) treatments were performed as a fresh media change. 0.5 µM tetrodotoxin (TTX; Tocris, UK) treatments were performed on day 3 of the recordings at CT2-3, administered as a microdrop to the existing media. Wash out of the TTX was performed as a complete media change at CT2-3 on day 7 Control treatments were made analogously, with a microdrop of water.

*Image analysis—manual tracking*: Images were primarily analysed in ImageJ, using a region of interest tool to select putative single cells or whole brain areas for assessing relative bioluminescence over time: AP, NTS, and 4thVep for baseline recordings and TTX treatments and separately NTSc and NTSd (connected and disconnected from the AP, respectively) for the mechanical disconnection protocol. The first 12 h of all recordings were excluded and raw data were subject to a 3 h running average smooth before further analysis. Peaks and troughs of individual bioluminescence traces of whole structures and putative single cells were determined manually. An average of three peak measurements were used to measure period. Damping rate was determined as relative to the amplitude of the peak on day 2. To determine the spatiotemporal amplitude heatmap for the whole DVC, the OriginPro 9.1 (OriginLab, USA) software was used. Rayleigh plots of peak phase and subsequent R values were created using El Temps (University of Barcelona, Spain). For these plots, triangles represent peak bioluminescent phases for individual cells, the arrow indicates the phase vector and its length represents significance with respected to the inner dotted circle of the $p = 0.05$ significance threshold. Variance is represented by the black box around arrow heads. Cross-correlograms indicating the differences in phasing were calculated in Statistica (StatSoft, USA).

*Grid analysis*: Time-lapse images were first treated with noise removal using Remove Outliers function in ImageJ (NIH, USA). The images were then stabilised using a set of ImageJ Plug-ins (Image Stabilizer and StackReg). From stabilised time-lapse images, spatiotemporal dynamics can be characterised by treating PER2::LUC oscillations in single-cell sized grids (5 × 5 pixels), analogous to voxels in brain imaging[28]. To maximize the chance that a grid represents a single-cell at each time point, cropped 410 × 210-pixel images were convoluted with a 5 × 5 homogeneous kernel. The images were subsequently reduced to matrices of 82 × 42 grids. Actively bioluminescent grids were selected in the order of high time average values ($n = 700$–900 grids).

*Oscillator phase and correlation analysis*: Each grid had oscillating temporal profile with varying amplitudes, and its phase was extracted by delay embedding (Takens' theorem). The period of its grid was estimated by the inverse of dominant frequency from Fast Fourier Transform (FFT) for the first 2 days in culture. This method tended to weigh higher on oscillations prior to damping. Amplitude-normalised oscillations were obtained by taking sine of extracted phases. Pair-wise correlations of the normalised oscillations were calculated against time-series of all grids, to make a correlation matrix and the partial correlation matrix was used to identify clusters[28]. This revealed spatial heterogeneity of oscillatory activities that matched with known anatomical areas such as AP and NTS.

*Detuning analysis*: The circadian oscillators in AP, NTS, and adjacent regions are initially well-synchronised and start to detune (desynchronise) over time in culture. When the variance of the initial frequency distribution is known, the detuning dynamics can be mathematically predicted for the uncoupled case (decoupling; coupling constant $K = 0$ in the Kuramoto model). It can be shown to follow a quadratic time course proportional to frequency variance[28].

$$Detuning \sim e^{-Dt^2/2}$$

where $D$ indicates frequency variance, which was estimated from initial 2-day's FFT, as described in the previous subsection.

*Statistics*: Statistical analyses were performed using Prism 7 (GraphPad Software, USA). To compare periods, standard deviation of periods (SD period) and time of peaks, paired *t* test and repeated measures (RM) one-way ANOVA followed by Tukey's multiple comparison were used. To assess the differences in values measured in the 3–4 subsequent days to asses damping and synchronisation (times of peaks, SD of single cell lags, Rayleigh R), RM two-way ANOVA followed by Sidak's multiple comparison was used, with the interaction value used to evaluate the difference in rate of the parameter change. The interaction in RM two-way ANOVA was also used to compare the shape of frequency histograms of single cell lags to the whole structure peak. Linear regression model was used to measure the desynchronisation rates. For detuning analysis, we used Welch's *t* test and paired *t* test using Mathematica (Wolfram Research, IL). All data was presented as mean ± standard error of mean (SEM). Both manual analysis and grid analysis yielded similar findings.

**Quantitative RT-qPCR.** *Tissue preparation*: A total of 19 male C57BL6J mice, aged 10 weeks old, were placed into constant darkness for 36–48 h before being culled at four time points across the circadian cycle: CT0, CT6, CT12, CT18 ($n = 4$–5). Brains were excised and flash frozen in dry ice before being stored at −80 °C for up to 4 weeks. Sections of 20 μm were cut using a cryostat (Leica CM3050 S), onto PEN-membrane slides (Leica Biosystems, Germany) and then stored at -80 °C for up to 2 weeks. For laser-capture microdissection, slides were first defrosted and then stained with 1% cresyl violet (Sigma) in 70% ethanol before regions of interest were extracted on laser-capture microscope (LCM) system (Leica DM6000 B) and stored in lysis buffer (Promega) prior to RNA extraction (see below). LCM performed on the cresyl violet-stained tissue enabled precise extraction of the desired areas (i.e. those which demonstrated the strongest bioluminescent signal measured in the PER2::LUC mice): (1) the AP with the surrounding glial border and (2) the bilateral medial NTS adjacent to the AP, excluding the DMV (Supplementary Fig. 9).

*RNA extraction for RT-qPCR*: The dissected tissue was then immediately processed for RNA extraction using the ReliaPrep RNA Tissue Miniprep System (Promega, USA) and subsequently stored at −80°C. Reverse-transcription was performed using the High-Capacity RNA-to-cDNA Kit (Applied Biosystems, USA) after which quantitative RT-PCR was performed using Power SYBR Green Master Mix and measured using a StepOnePlus Real Time PCR system (Life Technologies, USA). Transcripts were amplified by QuantiTect primer assay (Qiagen, Germany) for all genes (*Cckbr, Cldn5, Ghsr, Hcrtr1, Hcrtr2, Tbp* < housekeeping gene> and *Tjp1*) except the following primers for *Per2* (5′–3′: forward GCCTTCAGACTC ATGATGACAGA, reverse TTTGTGTGCCTCAGCTTTGG) and *Bmal1* (5′–3′: forward CCAAGAAAGTATGGACACAGACAAA, reverse GCATTCTTGATCC TTCCTTGGT).

*Data analysis and statistics*: The ΔΔCT method was used for data analysis with *Tbp* as the house keeping gene and CT0 values for relative target gene expression. One-way ANOVA and Kruskal-Wallis tests were performed using Prism 7 (GraphPad Software) for statistical analysis. Data was presented as mean ± SEM.

**Multi-electrode array electrophysiology.** *Tissue preparation*: In total, 39 male adult C57BL6J mice were culled at ZT0 (for long-term and acute early day recordings) or ZT11 (for acute early night recordings). The brain was carefully excised and immersed in the pre-oxygenated with carbogen (95% oxygen, 5% $CO_2$), ice-cold incubation artificial cerebrospinal fluid (ACSF) composed of (in mM): NaCl 95, KCl 1.8, $KH_2PO_4$ 1.2, $CaCl_2$ 0.5, $MgSO_4$ 7, $NaHCO_3$ 26, glucose 15, sucrose 50 and Phenol Red 0.005 mg/l. The brainstem was then separated and sliced using a vibroslicer (Camden Instruments). 250 μm (for acute set up) or 300 μm thick (for long-term recordings) coronal brain slices (at the corresponding stereotactic coordinates to those prepared from PER2::LUC mice), were cut and incubated in the cold incubation ACSF constantly bubbled with carbogen. After 30 min, slices were transferred to the oxygenated pre-incubation chamber filled with recording ACSF warmed up to 32°C, composed of (in mM): NaCl 127, KCl 1.8, $KH_2PO_4$ 1.2, $CaCl_2$ 2.4, $MgSO_4$ 1.3, $NaHCO_3$ 26, glucose 5, sucrose 10 and Phenol Red 0.005 mg/l.

*Recording and data acquisition*: Slices containing the DVC were then transferred to the recording wells (at projected ZT1 for the early day and long-term recordings, or ZT13 for early night recordings) of the MEA2100-System (Multichannel Systems GmbH, Germany) and placed onto the 6 × 10 perforated multi-electrode arrays (MEAs; 60pMEA100/30iR-Ti, Multichannel Systems). Slices were perfused with fresh recording ACSF, preheated to 32 °C with a perfusion cannula (Multichannel Systems), throughout the whole experiment. The tissue was allowed to settle for 1 h following set up before starting recording. For long term recordings that lasted more than 24 h, suction was set to minimum and ACSF was enriched with 1% Penicillin-Streptomycin (Sigma) to maximize the survival of the tissue. In order to verify the tissue condition after over 24 hours of recording, 100 μM NMDA (Tocris Bioscience, UK) was applied to provide evidence of tissue responsiveness. After short-term recordings, 1 μM TTX (Tocris) was applied to verify that the registered signal comprised of TTX-sensitive action potentials and to assess the signal-to-noise ratio.

The raw signal was sampled at 25 kHz and acquired using MC Rack 4.6.2 Software (Multichannel Systems). For spike detection, MC DataTool 2.6.15 (MultiChannel Systems) and MC Rack were used. First, data was high pass filtered

in forward and reverse directions (300 Hz). Then, spikes were classified as the signal exceeding the threshold of -17.5 μV (twice as high as determined to be a noise level after TTX application). Spontaneous multi-unit activity (sMUA) was further analysed only from recording spots that were classified to be localised in the AP or NTS.

*Drugs*: All drugs: orexin A (OXA, 200 nM; Tocris), ghrelin (200 nM, Tocris) and cholecystokinin (CCK; 1 μM, Tocris) were stored as 100x concentrated and were freshly diluted in recording ACSF prior to the application. To expose studied structures to elevated glucose, 10 mM of sucrose in the recording ACSF was exchanged to equimolar concentration of glucose (final concentration at 15 mM). All drugs were delivered by bath perfusion.

*Stimulation*: Stimulations were performed by switching the mode of four chosen electrodes from recording to stimulation mode. Stimulation electrodes were localised in the middle of the AP, which was assessed visually and electrophysiologically as recording locations in the AP are characterised by distinctly higher MUA than those in the NTS. Thus the stimulation area was of a triangular shape, with three electrodes in one row and one in the row below (Fig. 3a). The stimulation protocol was built from trains of ten negative voltage pulses (amplitude: 200 mV, duration: 5 ms, inter-stimulus interval: 5 s). Each train was repeated three times every 2 min (a 75 s stimulation break; Fig. 3b).

*Data analysis and statistics*: For short-term recordings, baseline sMUA was averaged from 1800 s intervals in NeuroExplorer 5 (Nex Technologies, USA). Recording locations were classified as "active" if the sMUA exceeded 0.005 Hz. The proportion of "silent" sites in each structure between early day and early night was compared using Fisher's exact test and the sMUA frequency - with Mann-Whitney tests. For the long-term studies, daily sMUA profiles were calculated in 600 s bins. The time of peak activity was manually assessed for sMUAs for each recording electrode localised in the AP and NTS and further analysed in El Temps for phase clustering before comparison with unpaired *t* test. Activity heatmaps were plotted in OriginPro 9.1 (OriginLab) from averaged baseline sMUA values.

Drug responses were analysed on 1 s binned MUA histograms in Spike2 8.11 (Cambridge Electronic Design Ltd., UK) and further in MatLab R2013a (MathWorks, USA) with custom-written scripts. The change in MUA was classified as a response if it exceeded 3 standard deviations (SDs) from the baseline. The maximal amplitudes of the responses were compared between day and night using unpaired *t* tests or Mann-Whitney tests, and the relative proportion of responsive spots with Fisher's exact tests. Responses were relicated across brain slices prepared from different animals.

Responses to electrical stimulation were prepared for analysis in Spike2 8.11 (Cambridge Electronic Design Ltd., UK). First, waveform data was transformed with the use of kernel function to generate Gaussian probabilities of spike generation over time (spike density function). Next, modified peri-stimulus time histograms (PSTHs) were generated, based on the sum of spike density functions around all 30 stimuli. Further analyses were performed with custom-made MatLab script (MatLab R2013a, MathWorks, USA). Each recording location in the NTS was classified as responsive to the AP stimulation, if the change in spike density function at the PSTH exceeded one standard deviation (SD) from the baseline mean. The amplitudes of responses were compared using Mann-Whitney test and the excitation-to-inhibition ratio between day and night recordings were analysed using Fisher's exact test. All responses localised immediately adjacent and subjacent to the stimulation electrodes were excluded from further analysis, due to the possibility of a passive response. Responses were relicated across brain slices prepared from different animals.

Statistical analyses were performed in Prisms 7 (GraphPad) and all data were presented as mean ± SEM.

**Evans Blue and immuno-histochemistry.** *Tissue preparation*: A total of 20 male C57BL6J mice, aged 10 weeks old, 11 of which were placed in constant darkness (DD) for 36–48 h prior to cull. 4 groups of mice were subjected to transcardial perfusions: $n = 5$ at ZT1, $n = 4$ at ZT13, $n = 5$ at CT1 and $n = 6$ at CT13. After terminal injection of pentobarbital (80 mg/kg), transcardial perfusions were performed with ice-cold phosphate-buffered saline (PBS) (flow rate: 7 ml/min, volume: 35 ml) followed by a 1% Evans Blue dye (EB, Sigma) and 4% PFA in PBS cocktail (flow rate: 7 ml/min, volume: 50 ml). Brains were subsequently excised and immersed in 4% PFA in PBS to post-fix overnight at 4 °C, after which they were immersed in 30% w/v sucrose solution for 2-3 days. Fixed hindbrains were separated before mounting on a freezing sledge microtome (Bright Instruments, UK) and cut into 35 μm thick to collect sensory circumventricular organs: the vascular organ of the lamina terminalis (OVLT), the subfornical organ (SFO) and the DVC.

*Immunohistochemical staining*: Slices containing the ROIs were rinsed off in PBS and permeabilised with 0.1% Triton X-100 (Sigma) for 20 min at room temperature (RT). Following washing in PBS, sections were blocked in 5% normal donkey serum (NDS; Sigma) with 0.05% Triton X100 for the next 30 min at RT. Then, slices were transferred to a PBS solution containing 0.5% NDS, 0.05% Triton X100 and the primary antisera: chicken anti-vimentin (1:4000, Abcam, Cambridge, UK), and mouse anti-GFAP (1:400, sigma) for 48 h at 4°C. Sections were subsequently washed in PBS and placed in the PBS solution of anti-chicken Alexa488 (1:800, Jackson ImmunoResearch, West Groove, USA) and anti-mouse Cy3 (1:800, Jackson ImmunoResearch) conjugated secondary antibodies raised in donkey. After 24 h at 4 °C, immunostained slices were rinsed off in fresh PBS and

mounted on gelatine-coated glass slides with VectaShield medium (Vector Laboratories, USA). Details on antibodies used in this protocol are presented in Supplementary Table 9.

*Data collection, analysis and statistics*: Three brainstem sections (caudal AP, intermediate AP, rostral AP), one mid-level SFO and one mid-level OVLT section per mouse were imaged on a confocal microscope (Leica SP5 upright) under the 20x objective. Florescence emitted by EB was detected at 650 nm. Z-stacks were collected with 3 μm steps and analysed as maximal intensity projections. Further analysis was performed in ImageJ. As no EB signal was observed outside SFO or OVLT in any of the experimental groups (Supplementary Fig. 10), detailed image analysis was performed for brainstem sections only. Maximal intensity projections were created for each stack to analyse the intensity of EB-derived signal in the NTS. Regions of interest (ROI) were anatomically distinguished using vimentin-immunoreactivity (ir) to delineate the border between the AP and NTS and GFAP-ir was used to establish ventral border of the DMV and the lateral border of the NTS. Subsequently, the area of NTS (*whole*) was divided into two ROIs: *medial* and *lateral*, by halving the ventral and dorsal border of the NTS with a linear vector (Fig. 5a, top panel). The intensity of each ROI (mean grey value) was reduced by the background value, collected with the $40 \times 40$-pixel oval at the area ventral to DVC (which lacked EB staining). The intensity values for 3 ROIs were compared between corresponding slices in ZT/CT1 vs ZT/CT13, using unpaired *t* tests or Mann-Whitney tests in Prism 7 (GraphPad). Data were presented as mean ± SEM. Results obtained under LD were replicated under DD conditions.

**Reporting Summary**. Further information on research design is available in the Nature Research Reporting Summary linked to this article.

## Data avaliability

The source data used to generate the charts within the manuscript is available as supplementary data 1. Data that support the findings of this study is available from the corresponding author upon reasonable request.

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

## Acknowledgements

This work was supported by a project grant BB/M02329X from the BBSRC (UK) to HDP and a MRC DTP Studentship to RCN. LC was additionally supported by a Polish National Science Centre doctoral scholarship "Etiuda IV" 2016/20/T/NZ4/00273. JM was supported by the Taiwan Ministry of Science and Technology (107-2311-B-038-001-MY2, 107-2410-H-038-004-MY2, 108-2321-B006-023-MY2, 108-2410-H-038-008-MY2), the Higher Education Sprout Project by the Ministry of Education (MOE) in Taiwan, Taipei Medical University (TMU107-AE1-B15, 107-3805-003-110), Taipei Medical University-Shuang Ho Hospital (107TMU-SHH-03), and Nakayama Foundation for Human Science. We would like to thank Tomasz Blasiak, PhD (Jagiellonian University in Krakow, Poland) for sharing his custom-made MatLab scripts and expertise in the analysis of electrophysiological data. Drs. Ashleigh Wilcox, David Lyons, and Giuseppe D'Agostino provided valuable comments on earlier drafts of the manuscript. We thank the staff of the University of Manchester Biological Services Facility and Bioimaging Facility for their assistance.

## Author contributions

L.C., R.C.N. and H.D.P. conceived the project and designed experimental protocols. R.C.N. and L.C. performed and analysed bioluminescence imaging experiments. J.M. computed, interpreted grid-based analysis of bioluminescence imaging data, and performed simulation. L.C. carried out, analysed and interpreted electrophysiological studies. R.C.N. and L.C. designed and performed permeability studies, immunohistochemical staining and confocal imaging. L.C. and R.C.N. performed RNA extraction with the help of C.P. R.C.N., P.C. and C.P. designed, performed and interpreted qPCR experiments. L.C., R.C.N., H.D.P. and J.M. wrote the manuscript and all co-authors agreed to the final version. H.D.P. supervised the project and provided financial support for the study.

## Competing interests

The authors declare no competing interests.
