## [Peer Review File · Communications Biology]

Editorial Note: Parts of this Peer Review File have been redacted as indicated to maintain the confidentiality of unpublished data

Reviewers' comments:

Reviewer #1 (Remarks to the Author):

The authors provide evidence that in the dorsal vagal complex (DVC) circadian rhythms of a Per2:luciferase reporter construct in the area postrema (AE) and the nucleus of the solitary tract (NTS) can be observed. They also give a nice description of the coupling relationship between the AE and the NTS and draw a parallel between this coupling and the coupling of the subdivisions in the SCN using a computational approach. The rhythms observed in the AE and the NTS are paralleled by day/night changes in neuronal activity in these structures. Interestingly, the dye Evans blue (EB), when injected into the AP, diffuses out from the AP at ZT13 but not at ZT1 implying a time of day dependent permeability out of the AP for this dye. This also correlates with day/night effects of metabolites such as CCK, glucose, and the peptides ghrelin and orexin A on MUA activity. Furthermore, the expression of the corresponding receptors in the NTS display a circadian rhythm in mRNA expression. From these data the authors conclude that the DVC is a multi-oscillatory centre, where circadian mechanisms exert a prominent influence on molecular and cellular activity and function (line 440-441).

Major concern:

The authors show correlations between all the factors investigated but no causal relationships. For example, how would the investigated parameters look like if there were a destroyed circadian clock (e.g. Bmal1 ko animals). Another approach to show causality would be to knock-down a clock gene specifically in the AP or NTS and show how the investigated readouts are affected.

As the manuscript stands the observations all correlate well with the authors statement but there are no experiments that challenge their conclusions.

Reviewer #2 (Remarks to the Author):

Chorobok et al use multiple experimental approaches, both in vivo and ex-vivo to characterize circadian oscillators in subregions of the hindbrain including the area postrema and nucleus of the solitary tract, and elucidate a potential physiological function of these tissue-specific oscillations. This is a very elegant and important study, quite unique in terms of its multiple and complementary experimental approaches. The paper is well written, the experiments follow each other nicely, the methods are well described, data analyses are appropriate, and the conclusions are well stated and conservative. Overall, the findings of the paper are novel and add nicely to other literature on the presence and physiological/behavioral function of autonomous circadian oscillators in brain regions in mammals.

My only question relates to the computational modeling and simulation component of the paper. I see no reason to include this analysis in the paper. In my opinion, it adds little to the hard and very convincing data reported!

We thank the reviewers for their expert comments and suggestions on our manuscript, 'Timekeeping in the hindbrain: a multi-oscillatory circadian centre in the mouse dorsal vagal complex. We have attempted to address the points raised through removing the modelling section and through inclusion of a paragraph on strategies for examining causality. Unfortunately, international events have overtaken us, and our animal houses are shutdown, our other labs are about to be shutdown, and we no longer make or accept new deliveries of animals, reagents, etc. Further, key staff are self-isolating and/or working from home. This Covid-19 situation is likely to continue for another 8-12 weeks and possibly longer. Even then, it will take us a long time to get back up and running. Therefore we cannot in the short to medium term obtain mouse models and reagents to explore causality. This is the case for the UK, countries in the EU including Poland, as well as Asia including Taiwan. We do believe though that they have indications of causality from our existing datasets (as outlined below).

Reviewer 1

We thank the reviewer for his/her positive comments on this study. We agree that mechanisms of causality will be important to explore and address in future studies. Unfortunately, with the current crisis with Covid-19, we are unable to import new mouse strains (such as *Cry1*^{-/-}*Cry2*^{-/-} or *Bmal1*^{flox/flox}) or even viral vectors into our facilities. Across the UK, many EU countries, and Taiwan, movement of mouse lines is not permitted and in Bristol and Manchester, we are not allowed to begin new experiments for at least the next 8-12 weeks. Essentially, we have now wound down our research and several key staff are either self-isolating or working exclusively from home. These unprecedented circumstances have stopped almost all neuroscience and physiology research. Had Reviewer 3 executed his/her duties in a timely manner, then that might have allowed us sufficient time to obtain the appropriate tools to further explore causality of rhythms in the brainstem.

Based on our previous research (Sakhi *et al.*, *J. Physiol.* 2014a,b), we would be very surprised if the absence of a functional molecular clock did not suppress or abolish the daily/circadian rhythms in DVC neuronal activity shown in this study. The absence of the *Cry1* and *Cry2* genes abolishes the SCN molecular clock (van der Horst *et al.*, *Nature* 398:627-630, 1999). In our earlier published work on the brain's epithalamus—a structure which exhibits rhythms in clock gene expression independent of the SCN, we have shown that in both medial (Sakhi *et al.*, *J. Physiol.* 592: 587-603, 2014a) and lateral (Sakhi *et al.*, *J. Physiol.* 592: 5025–5045, 2014b) habenula, the absence of *Cry1* and *Cry2* abolishes day-night rhythms in neuronal activity as well as circadian rhythms in a reporter of the molecular clock, *Per1*-luc. Thus, unless DVC rhythms in neuronal activity are uniquely independent of the molecular clock, we would predict their absence would parallel the absence in local expression of key molecular clock components.

Technically, selectively targeting cells that express a clock gene in a specified structure of the DVC would be an ideal but very challenging strategy to assess causality. For pinpointing the NTS with viral vectors, this would involve novel surgical approaches to avoid off-target transduction of the AP (this would compromise any straightforward interpretation of the results). These approaches are feasible, but will take several months if not a year or two. As we are unable to even initiate such research, all we can do is acknowledge the importance of the causality question and the approaches to address it in a new paragraph in the Discussion.

It should be noted though that we have addressed some aspects of causality. We know that action-potential dependent communication is critical for the function of the SCN molecular clock and we show here that impairment of action potential-dependent communication with tetrodotoxin (TTX) also compromises molecular rhythms in the brainstem. Further, we show that severing anatomical

connections and hence communication between the AP and NTS alters some rhythmic properties of the NTS molecular clock. Thus, molecular rhythms in the NTS are shaped by input from the AP.

Reviewer 2

This reviewer is very positive about the study but suggests removing the computational model. We have done that and altered the text appropriately.

REVIEWERS' COMMENTS:

Reviewer #1 (Remarks to the Author):

I can understand the current situation about including mutant mouse data with new experiments. The authors acknowledge that their study provides the ground for future studies using mouse mutants in later studies to show causal relationships. This is mentioned now in the discussion.

Reviewer #2 (Remarks to the Author):

Well done!